# Histopathological Changes and Immune Response Profile in the Brains of Non-Human Primates Naturally Infected with Yellow Fever Virus

**DOI:** 10.3390/v17030386

**Published:** 2025-03-07

**Authors:** Suzana Ribeiro de Melo Oliveira, Ermelinda do Rosário Moutinho da Cruz, Nelielma Garcia de Oliveira Prestes, Fábio Silva da Silva, Marialva Tereza Ferreira de Araújo, Orlando Pereira Amador Neto, Maria de Lourdes Gomes Lima, Bianca Nascimento de Alcântara, Daniel Damous Dias, Jorge Rodrigues de Sousa, Arnaldo Jorge Martins Filho, Livia Medeiros Neves Casseb, Daniele Barbosa de Almeida Medeiros

**Affiliations:** 1Department of Arbovirology and Hemorrhagic Fevers, Evandro Chagas Institute, Rodovia BR-316 km 7 s/n—Levilândia, Ananindeua 67030-000, Pará, Brazil; lan.suzanaribeiro@gmail.com (S.R.d.M.O.); nelielmaprestes@gmail.com (N.G.d.O.P.); bicalcantara@gmail.com (B.N.d.A.); damous1994@gmail.com (D.D.D.); krekrodrigues@gmail.com (J.R.d.S.); dbamedeiros@gmail.com (D.B.d.A.M.); 2Department of Clinical and Experimental Pathology, Evandro Chagas Institute, Rodovia BR-316 km 7 s/n—Levilândia, Ananindeua 67030-000, Pará, Brazil; ermelinda_mc@yahoo.com.br (E.d.R.M.d.C.); marialvaaraujo@yahoo.om.br (M.T.F.d.A.); orlandoneto@iec.gov.br (O.P.A.N.); lourdes_lima59@hotmail.com (M.d.L.G.L.); arnaldofilho@iec.gov.br (A.J.M.F.)

**Keywords:** yellow fever virus, natural infection, central nervous system

## Abstract

In the history of yellow fever (YF) outbreaks in Brazil, howler monkeys (*Alouatta* sp.) and marmosets (*Callithrix* sp.) have been among the most affected genera, exhibiting significant hepatic injuries similar to those seen in humans. However, limited information exists regarding yellow fever virus (YFV) infection in their central nervous system (CNS). To address this gap, an epidemiological study was conducted to assess tissue changes, viral detection, and cytokine profiles in the brains of both neotropical primate species when they are naturally infected with YFV. A total of 22 brain samples from these species (8 from *Alouatta* sp. and 14 from *Callithrix* sp.) showing infection with YFV in the liver via immunohistochemistry (IHC) were selected. From them, YFV antigen detection occurred in 35.7% (5/14) of *Callithrix* sp. brain samples and 87.5% (7/8) of *Alouatta* sp. samples, with a higher frequency of viral antigen quantification in *Callithrix* sp. Both species exhibited similar CNS lesions, characterized by congestion, low hemorrhage, limited inflammatory infiltration interstitial and perivascular edema associated with neuronal degeneration, neurophagy, and higher cell death (necrosis and apoptosis) quantification. Pro- and anti-inflammatory cytokine profiles were balanced, with TNF-α and IL-1β playing a key role in inflammation, while IL-10 and IL-13 exhibited a prominent role in immunomodulation, suggesting an anti-inflammatory modulation typical of flaviviruses occurs. This study demonstrates that YFV can induce CNS lesions in neotropical primates, establishing it as a secondary target of viral tropism. These findings highlight the importance of collecting nervous tissue during epizootics, particularly in *Callithrix* sp., as such tissue is often overlooked despite its critical role in disease monitoring.

## 1. Introduction

Yellow fever (YF) is an infectious disease caused by yellow fever virus (*Orthoflavivirus flavi*, YFV)—family *Flaviviridae*, genus *Orthoflavivirus*—leading to a febrile case until a severe ictero-hemorrhagic condition, wherein 10% of confirmed case progress to death [1]. As an arthropod-borne virus, YFV has two distinct maintenance cycles: the urban cycle, in which man is the host and the main vector is *Aedes aegypti*, and the sylvatic cycle, where the virus is kept in nature among non-human primates (NHPs) and mainly mosquitoes of the genus *Haemagogus* sp. and *Sabethes* sp. [2].

The sylvatic cycle has been related to epidemics and enzootic outbreaks in the rain-forest areas of South America [3]. In Brazil, the majority of YF outbreaks have been reported in the north (Amazon region) and central-west regions, over an interval of 6–10 years. However, from 1999 onwards, YFV spread to non-endemic regions, affecting a large number of humans and NHPs in the southeast and south of Brazil [4]. From 2015–2018, the largest YF outbreak reported in the last 80 years occurred, affecting areas outside the endemic regions, including in some of the most populous regions of the country (Minas Gerais, Espírito Santo, Rio de Janeiro, and São Paulo States). This highlighted concerns about the risks of the re-urbanization of YF [5,6].

In this type of epidemiological scenario, epizootic surveillance has played a crucial role in initiating preventive actions, particularly in the immediate vaccination of susceptible human populations [7]. During the most recent YF outbreak, approximately 2400 NHPs (including *Callithrix* sp., *Alouatta* sp., *Sapajus* sp., and *Callicebus* sp.) were affected, originating from 21 of Brazil’s 27 states. Although Neotropical NHPs are clearly susceptible to YFV, the prevalence and viral load can vary depending on the genus [8,9].

Howler monkeys (genus *Alouatta*) are considered to be sentinels of YFV circulation. Even with low viral loads, infection with YFV results in a high fever, prostration, jaundice, anorexia, oral and intestinal hemorrhage as well as damage to the liver, kidneys, pancreas, spleen, heart, bone marrow, and CNS, which occur between the third to seventh days after the first episode of fever. Those infected may develop a fatal hepatic failure with cell death, occurring between the third to seventh day after the first episode of fever [9,10].

*Alouatta* sp. is a NHP model used to study YF pathogenesis due to the fact that their hepatic changes are similar to those found in humans—namely, high levels of necrosis/apoptosis associated with Councilman–Rocha Lima bodies, steatosis, and mild inflammatory infiltrates [9,11,12,13]. These lesions are related to both viral replication on hepatocytes and the primary cellular responses of *TCD4^+^* lymphocytes followed by the actions of *TCD8^+^* lymphocytes, macrophages, polymorphonuclear cells, natural killer cells (NK), complement systems, and cytokines (namely, *TGF-β*, *TNF-α,* and *IFN-γ*) [12]. The secondary systemic reaction to infection with YFV may lead to excessive congestion, edema, hemorrhage, and mononuclear inflammatory infiltrate [14,15].

*Callicebus* sp. and *Sapajus* sp. have shown a susceptibility to YFV and have developed a full spectrum of liver lesions, similar to those in *Alouatta* sp. and humans, but with a high viral load [9]. Marmosets (*Callithrix* sp.) naturally infected with YFV have been shown to have a low prevalence for YF and typical hepatic lesions have not been observed [16]. The susceptibility to YFV and hepatotropism of callitrichids has also been confirmed in experimental studies [17]. Other species, such as *Saimiri*, *Saguinus*, *Aotus*, *Ateles*, *Cebus*, and *Sapajus* genera, have been considered less susceptible to YFV, even when infected with a high viral load. These species hardly develop serious disease, presenting when infected with a fast febrile picture and short viremia [8,18,19]; however, experimental studies have demonstrated that *Saimiri*, *Saguinus,* and *Sapajus* specimens may develop the disease with fatal outcomes [20,21].

Despite YFV being commonly associated with systemic and hemorrhagic clinical syndromes, neurological manifestations have also been reported [22]. Although rare, YFV immunization can occasionally induce encephalitis, which, while typically controlled by the host, remains a noteworthy clinical concern [23,24,25]. Given this, we hypothesized that different NHP species may exhibit variations in YFV organ tropism or viral replication, and that relying exclusively on liver samples might limit the effectiveness of YF surveillance and monitoring. To explore this, we analyzed histopathological samples from *Alouatta* sp. and *Callithrix* sp., the two most frequently studied genera for YF diagnosis during the most recent YF outbreak investigated by the Evandro Chagas Institute.

To reduce this knowledge gap, we characterized and compared the histopathologic signature of YF-associated liver disease, viral antigen detection by immunohistochemical (IHC) analysis, and molecular findings in samples from humans and NHPs infected with YFV that were received from 2015 to 2017 at the Adolfo Lutz Institute (São Paulo, Brazil).

## 2. Materials and Methods

### 2.1. NHP Samples

To carry out this study, 25 paraffinized samples from NHPs were selected from collections available in the Pathology Section at the Instituto Evandro Chagas (SAPAT/IEC) from investigations into epizootic diseases in the states of Minas Gerais, Goiás, Rio de Janeiro, and the Federal District from 2015 to 2017. Of these, 22 paraffinized brain samples belonging to the genera *Callithrix* sp. and *Alouatta* sp. had a positive diagnosis for YFV, which was confirmed via IHC testing of the liver tissue samples and via the presence of significant brain lesions. Furthermore, these animals also had lesions suggestive of encephalitis. Another four (03) animals were selected as a negative control group because they presented with negative results for YFV and showed an absence of brain tissue damage (Appendix A).

### 2.2. Histopathology

The organ and viscera fragments of the NHPs were directed to the SAPAT/IEC, preserved in 10% buffered formalin and were analyzed macroscopically and submitted to cuts of approximately 3 mm^3^. Subsequently, these samples were processed for routine histopathological analysis, using a Leica ASP300S, according to the following sequence: three baths in 70%, 80%, and 95% alcohol, four baths in absolute alcohol, three baths in xylene, and three baths in paraffin. At the end of this processing, the samples were embedded in paraffin blocks with the aid of HistoCore Arcadia equipment. The blocks were sectioned using a micrometer (Microm HM 340E) to obtain 4 μm-thick slices, which were mounted on glass slides for histological analysis. The tissue sections were then stained using the routine Hematoxylin and Eosin (HE) technique [26]. The stained slides were examined under optical microscopy (Zeiss, Oberkochen, Germany) at 100× and 400× magnifications by two independent pathologists, focusing on regions proximal to the meninges, parenchyma, and perivascular areas.

### 2.3. Immunohistochemistry for YFV Antigen Detection

Of the 22 NHPs with a positive diagnosis for YFV, the slides equivalent to the brain were separated. These were submitted to the immunohistochemistry (IHC) technique via the alkaline phosphatase method, using polyclonal antibodies specific for YFV that were produced in the Department of Arbovirology and Hemorrhagic Fevers (SAARB/IEC) [10].

After deparaffinization with xylene and hydration with decreasing series of alcohol baths (absolute I, absolute II, 95%, 80% and 70%), the slides were washed with distilled water and then placed in an incubator solution at 37 °C. Subsequently, antigenic recovery was performed in a protease solution (Sigma-Aldrich, Poole, UK) at 37 °C. Non-specific proteins were blocked with a solution of 5% skimmed milk, 0.01 M phosphate-buffered saline (PBS), and normal horse serum (Vector Laboratories), with an incubation period of 20 min, at room temperature (RT).

Incubation was performed with the primary antibody (Anti-YFV) at a 1:200 dilution in PBS (pH 7.2) for 1 h at RT. After three washes with PBS, the slides were treated with a biotinylated anti-mouse IgG secondary antibody (Vector Laboratories), diluted 1:5 and incubated for 1 h at RT. Next, the slides were incubated with a streptavidin–alkaline phosphatase complex (Invitrogen, Waltham, MA, USA) at a 1:20 dilution for 1 h at RT, followed by three washes with a 0.1 M TRIS solution. Subsequently, a HistoMark Red Phosphatase Kit was applied to the slides for 30 min at RT to detect antibody staining. The sections were then counterstained with Harris hematoxylin, and the slides were mounted with Entellan (Merck Millipore, Burlington, MA, USA). Finally, the slides were analyzed under an AXIO IMAGER Z1-ZEISS optical microscope (model 4560006, ZEISS, New York, USA) using 10× and 40× objective lenses by a physician and veterinary pathologists.

### 2.4. Immunohistochemistry for the Immune Response

To study the tissue immune response, pro-inflammatory (TNF-α, IFN-γ, IL-1β, and IL-12) and anti-inflammatory (IL-4, IL-10, IL-13, IFN-β, and TGF-β) cytokines as well as cell death apoptosis biomarkers (Appendix A), using IHC with the biotin–streptavidin peroxidase complex, were used. These cytokines were selected to characterize the Th1/Th2 immune response profile. Th1-associated cytokines (e.g., TNF-α, IFN-γ, and IL-12) are involved in pro-inflammatory responses, which may contribute to neuronal damage, while Th2-associated cytokines (e.g., IL-4, IL-10, and IL-13) play a regulatory role in limiting inflammation. Additionally, TGF-β and IFN-β are critical modulators of immune responses in the CNS, influencing neuroinflammation and tissue repair. Understanding these profiles helps elucidate the tissue response induced by YFV infection in these neotropical primates.

Initially, tissue samples on slides were deparaffinized in 100% xylene, twice, for 15 and 10 min, and hydrated in a decreasing series of ethyl alcohol—absolute I and absolute II for 5 and 3 min, respectively, and in alcohol at 95%, 80%, and 70% for 2 min each step. Then, the slides were washed in PBS and distilled water for 5 min each. Endogenous peroxidase was blocked with 3% hydrogen peroxide (H_2_O_2_) in three (03) incubations of 15 min each, in a dark chamber at RT, followed by washes in PBS pH 7.2 and distilled water.

The antigenic recovery step was carried out with citrate buffer pH 6.0 plus Tween20 for 20 min at 90 °C in a water bath. Then, the slides were left to dry in RT for 10 min. The next step was to block nonspecific proteins, which was carried out via incubation in a 10% skimmed milk solution for 30 min at RT.

The histological sections were incubated with the primary antibodies diluted in a PBS solution containing 1% bovine albumin (BSA) for 14 h in a humid chamber at 4 °C. After this interval, the slides were immersed in a PBS buffer solution and then incubated with the biotinylated secondary antibody LSAB (DakoCytomation) in an oven for 30 min at 37 °C. After the first incubation, the slides were again immersed in PBS and incubated with streptavidin peroxidase (LSAB DakoCytomation) for 30 min at 37 °C. After this interval, the sections were revealed with the application of a chromogen solution composed of 0.03% diaminobenzidine and 3% hydrogen peroxide. At the end of the development step, the preparations were washed in distilled water and then counterstained with Harris hematoxylin for 1 min. Finally, the histological sections were dehydrated in ethyl alcohol in increasing concentrations and cleared in xylene.

### 2.5. Quantitative Analysis and Photodocumentation

Photos taken from slides were analyzed using the an AXIO IMAGER Z1-ZEISS field microscope (model 4560006). Immunomarkers were quantified by randomly selecting three microscopic fields within each of the following regions: meninges, perivascular, and parenchyma. Each IHC field was subdivided into 10 × 10 areas delimited by the graticule comprising a region of 0.0625 mm^2^, then the average was taken, and the value obtained was multiplied by 0.0625 comprising the total area. Results were stored in Microsoft Excel 2010 spreadsheets.

### 2.6. Statistical Analysis

The normality of the variables was assessed using the Shapiro–Wilk test, while the homogeneity of variances was verified using Levene’s test. Outlier detection was performed through boxplot analysis and the interquartile range (IQR) method.

To compare the medians of cytokine expression and lesion quantification between groups, the Wilcoxon rank-sum test with continuity correction (using the Mann–Whitney test) was applied. The comparison of YFV antigen expression among the studied regions within each group was performed using the non-parametric Kruskal–Wallis analysis of variance, followed by Dunn’s post hoc test with the Bonferroni correction for multiple comparisons. Correlations in cytokine expression were evaluated using Spearman’s test.

All analyses were conducted using the R software (version 4.4.0) [27]. Data processing was performed with the dplyr and reshape packages from the Tidyverse library [28]. The Shapiro–Wilk and Mann–Whitney tests were executed with the stats package [27], while Levene’s test was applied using the rstatix package [29]. Correlation matrices and correlograms were generated using the Hmisc [30] and corrplot [31] libraries, respectively.

A 95% confidence interval was adopted for all statistical analyses. Statistical significance was classified according to the following criteria: *p* > 0.05 (not significant, N.S.), *p* < 0.05 (significant), *p* < 0.005 (highly significant), and *p* < 0.0005 (extremely significant).

## 3. Results

A total of 22 non-human primates (NHPs) positive for YFV antigen detection via IHC in the liver, comprising 14 (63.6%) *Callithrix* sp. and 8 (36.4%) *Alouatta* sp, were selected. All of them were positive for YFV detection in the liver, confirmed via by IHC (Figure 1A).

Histopathological analysis revealed a range of hepatic lesions, including degenerative changes such as hepatocyte tumefaction, observed in 100% of individuals from both species. Steatosis (Figure 1C and Figure 2F) was more frequent in *Alouatta* sp. (75%) compared to *Callithrix* sp. (35.7%). Biliary disorders, such as cholestasis, were slightly more frequent in *Callithrix* sp. (42.90%) than in *Alouatta* sp. (25%). Cell death was evidenced by necrosis and apoptotic acidophilic Councilman bodies, highlighting that necrosis was more frequent in *Alouatta* sp. (100%) (Figure 1A).

Vascular alterations such as congestion were present in both species. Inflammatory changes, primarily perivascular infiltrates, were widely observed in both *Alouatta* sp. (75%) and *Callithrix* sp. (78.6%). Hemorrhage (Figure 1G) was less common, affecting only 12.5% of *Alouatta* sp. and 21.4% of *Callithrix* sp. Reparative changes, such as Kupffer cell hyperplasia and hypertrophy, as well as reactive hepatitis, were balanced in both species (Figure 1A; Appendix A).

Three regions of the CNS were analyzed: the meninges, cerebral parenchyma, and perivascular areas. Among the 22 monkeys, only 12 (12/22; 54.5%) showed YFV antigen detection via IHC in the brain, including 5 Callithrix sp. (5/14; 35.7%) and 7 *Alouatta* sp. (7/8; 87.5%) (Figure 2A). All animals that were negative for YFV detection in the brain also did not show significant lesions in this organ (Appendix A).

Regardless of the NHP species, the main histopathological findings in the meninges were edema (100%), neuronal degeneration (100%), cell death—apoptosis/necrosis (100%), and congestion (85.7–100%) (Figure 2F,I). Among individual NHPs, hemorrhage (60%) and inflammatory infiltrates (60%) had a higher frequency in *Callithrix* sp. (Figure 2B).

In the lesion quantification analysis, the severity of the various lesions found in meninge, parenchyma, and perivascular areas showed that *Callithrix* sp. did not exhibit significantly greater cerebral involvement compared to *Alouatta* sp. (Appendix A). In both species, the main meningeal lesions found were congestion, inflammation, perivascular edema (Figure 2F), and hemorrhage. The medians (red dots) indicate a slightly higher tendency for inflammation, perivascular edema, and hemorrhage in *Callithrix* sp. (Figure 2L). In parenchyma, the main injuries observed were interstitial edema and cell death characterized by necrosis or apoptosis (Figure 2D,E). The *Callithrix* sp. showed a greater involvement (of cell death), as evidenced by the higher median (red dot) in this group. Neuronal degeneration, neuronophagia, and satellitosis were detected in both groups, but with a lower intensity and no statistically significant differences (Figure 2M). Interstitial hemorrhage was rare (Figure 2C,D). Perivascular involvement included congestion and perivascular edema (Figure 2F), inflammatory infiltrate, cell death (Figure 2G), and vascular injury, which was occasionally accompanied by hemorrhage. The median suggests a slight tendency toward greater inflammatory injury and cell death in *Callithrix* spp. (Figure 2N).

Quantitative analysis of YFV antigen detection by IHC revealed that, for both species, the parenchyma had the highest number of YFV antigen-positive cells (median), indicating a statistically significant difference in meningeal (M) and perivascular (P) lesions areas (Mann–Whitney test, *p* < 0.05) (Figure 3A,B and Appendix A). Notably, in both the meninges (Figure 3F,I) and perivascular areas (Figure 3H,K), immunostaining was predominantly observed in endothelial cells. In the meninges, YFV antigen was also detected in mononuclear cells within the inflammatory infiltrate. In the neural parenchyma, antigen detection was primarily observed in neurons, followed by glial cells (Figure 3G,J). Interestingly, YFV antigen detection was more prominent in *Callithrix* sp. than in *Alouatta* sp. in the meninges (Mann–Whitney test, *p* = 0.0062) and parenchyma (higher median level; *p* > 0,05). The average total of YFV antigen-positive cells in the brain was 328 positive cell/0.0625 mm^3^ for *Callithrix* sp. and 169 positive cell/0.0625 mm^3^ for *Alouatta* sp., suggesting that Callithrix sp. has a higher tendency to favor YFV replication in the CNS (Figure 3B).

### Profile of the Immune Response in the CNS

Quantification of pro-inflammatory cytokines (TNF-α, IFN-β, IFN-γ, IL-1β, and IL-12) and anti-inflammatory cytokines (TGF-β, IL-4, IL-10, and IL-13) revealed variations in levels between the *Alouatta* sp. and *Callithrix* groups, but no statistically significant differences were observed using the Mann–Whitney test (Appendix A). This lack of statistical difference suggests that, despite individual variations, both groups activate similar immune response patterns in all CNS areas, without a clear predominance of either an inflammatory or regulatory profile in either species (Figure 4S–P).

Among the pro-inflammatory cytokines, TNF-α (Figure 4D–F) and IL-1β (Figure 4G–I) exhibited higher median levels in both groups, indicating a central role in the inflammatory response to YFV. IFN-γ and IL-12, associated with cellular immunity activation, showed smaller variations, while IFN-β maintained more homogeneous levels, possibly reflecting stricter regulation during infection (Figure 4S–P).

In the anti-inflammatory profile, IL-13 (Figure 4J–L) and IL-10 demonstrated the highest median levels, suggesting a potential mechanism for modulating inflammation. Conversely, IL-4 and TGF-β (Figure 4M–O), linked to the Th2 response [32], exhibited lower relative expression (Figure 4S–P).

Regarding apoptosis immunomarkers (Figure 4P–R), our analysis revealed that both *Alouatta* sp. and *Callithrix* sp. species exhibited similar levels of caspase-3 detection. This suggests that, despite the observed variations in cytokine profiles, the apoptotic response mediated by caspase-3 is comparable between the two species (Figure 4S–P).

The correlograms revealed statistically significant correlations (*p* < 0.05) among YFV antigen-positive cells, cytokine expression, and apoptosis markers in *Alouatta* sp. and *Callithrix* species (Figure 5, Appendix A).

In *Alouatta* sp., TGF-β showed a strong positive correlation with the presence of the YFV antigen (ρ = 0.818; *p* = 0.0246) in parenchyma (Figure 5B), while TNF-α (ρ = −0.924; *p* = 0.0029; Figure 5C) and IL-13 (ρ = −0.882; *p* = 0.0086; Figure 5A) exhibited significant negative correlations, suggesting a potential regulatory role in viral persistence and immunomodulation. These immune markers were predominantly localized in the parenchyma, perivascular region, and meninges, respectively. Additionally, individual analyses indicated a significant negative correlation between IL-12 and TGF-β (ρ = −0.780; *p* = 0.0388; Figure 5C) in the perivascular region and between TNF-α and Caspase-3 (ρ = −0.830; *p* = 0.0208; Figure 5B) in the parenchyma, suggesting an inverse relationship between pro-inflammatory signaling and apoptotic activity.

In *Callithrix* sp., IL-10 was significantly upregulated in association with YFV antigen detection (ρ = 1.000; *p* < 0.0001; Figure 5D), emphasizing its possible immunosuppressive role in viral pathogenesis. Moreover, a positive correlation was observed between the pro-inflammatory cytokines IL-4 and IL-1β (ρ = 0.913; *p* = 0.0305) as well as between IL-4 and IL-13 (ρ = 0.913; *p* = 0.0305) in the perivascular region (Figure 5D), suggesting a coordinated immune response involved in tissue inflammation and potential viral clearance. TNF-α also showed a positive correlation with IL-10 (ρ = 0.917; *p* = 0.0285; Figure 5D) and IL-13 (ρ = 0.917; *p* = 0.0285; Figure 5E), while IL-13 exhibited negative correlations with IFN-α (ρ = −0.884; *p* = 0.0467) and IFN-β (ρ = −0.884; *p* = 0.0467) (Figure 5D), reinforcing a potential immunomodulatory effect of these cytokines in the context of YFV infection.

Our findings reveal distinct immune responses to YFV infection in *Alouatta* sp. and *Callithrix* sp. *Alouatta* sp. exhibited a stronger pro-inflammatory and apoptotic response, with TNF-α and IL-13 negatively correlating with the viral antigen, suggesting an inflammation-regulating role. In contrast, *Callithrix* sp. showed IL-10 upregulation, indicating immune suppression and potential viral persistence. The positive correlation between IL-4, IL-1β, and IL-13 suggests a coordinated pro-inflammatory response, while IL-13’s negative correlation with IFN-γ/β points to impaired antiviral activity. These results suggest that *Callithrix* sp. may be more susceptible to persistent infection, whereas *Alouatta* sp. relies on inflammatory and apoptotic pathways to control viral spread.

## 4. Discussion

Between late 2015 and 2018, Brazil experienced the largest outbreak of sylvatic yellow fever in the last 80 years, with a significant increase in human cases and deaths as well as epizootics among NHPs. The outbreak began in late 2015 in the north and, by November 2016, YFV had spread toward the east and south of the country. Initially, it affected the midwest region, and by 2017/2018 it had reached the southeast region, with a substantial number of cases reported in São Paulo and Rio de Janeiro, increasing the risk of urban reintroduction of the disease [33].

Since the beginning of the outbreak, numerous epizootics have been reported on by the Ministry of Health and in other studies, primarily affecting marmosets (*Callithrix* sp.) and howler monkeys (*Alouatta* sp.) [8]. Analyzing the viral dynamics, a study by [34] identified amino acid polymorphisms in highly conserved positions of non-structural proteins (NS3 and NS5) in 12 YFV samples from mosquitoes, humans, and NHPs. These mutations may have contributed to the virus’s ability to infect and spread, particularly in states with historically low vaccination coverage [35,36,37,38].

Most neotropical NHPs are considered sensitive to YFV, surveillance records have predominantly involved *Alouatta* sp. Between 1996 and June 2016, of the 2221 NHP deaths attributed to YFV in Brazil, *Alouatta* sp. has accounted for 85.0% of cases, followed by *Callithrix* sp. (8.3%), *Sapajus* sp. (0.8%), *Cebus* sp. (0.22%), and *Saimiri* sp. (0.05%) [39]. *Alouatta* sp. is regarded as the genus most susceptible to YFV infection, often developing severe and fatal diseases [39,40], even with exposure to low viral inoculums [4]. However, YFV antibodies have been detected in some *Alouatta* samples, indicating that some animals survive the infection [41,42,43].

Sick or deceased NHPs are typically collected, and organ fragments—such as the liver, spleen, kidney, lung, and brain—are sent to reference laboratories linked to the Brazilian Ministry of Health for yellow fever diagnosis. Laboratory tools, including immunohistochemical and histopathological analyses, allow for the detection of YFV antigens and the identification of histological alterations caused by the virus, respectively [36,37,38,39,40]. These methods also contribute to understanding the pathogenesis of the disease in comparison to human cases [44].

Although yellow fever has been recognized for centuries, its impact on NHPs remains poorly understood. Most studies on yellow fever pathogenesis are based on experimental models using NHPs species such as *Macaca mulatta* (rhesus monkeys) [45,46], *Alouatta* sp. [10], and *Saimiri* sp. [20], as well as hamsters [47]. However, these studies predominantly focus on the liver. An experimental study by Hudson (1928) [45], conducted in three stages, investigated yellow fever pathogenesis in rhesus monkeys and reported minimal alterations in the CNS. The findings were limited to mild congestion, considered insignificant lesions, with little variation compared to control animals.

Regarding natural infections by YFV in neotropical NHPs, Cunha et al. (2019) [8] demonstrated a greater susceptibility of *Alouatta* to YFV infection compared to other species, including *Callithrix*, which supports our findings. Although *Callithrix* samples were more numerous (14/22; 63.63%) compared to *Alouatta* samples (8/22; 36.36%), a higher positivity rate in the CNS was observed in *Alouatta* (87.5%; 7/8) compared to *Callithrix* (35.7%; 5/14), as evidenced by IHC tests (Figure 1A and Figure 2A). It is important to note that the structural complexity and cellular diversity of the brain are greater than those of the liver, and due to the nature of the samples collected through yellow fever surveillance investigations, it was not possible to sample distinct brain regions from these animals. This limitation may introduce bias into the analyses. Therefore, only experimental studies could provide a more comprehensive understanding of this result.

Cunha et al. (2019) [8] also evaluated histopathological analysis and RT-PCR of NHPs naturally infected with YFV and suggested that viral loads in *Callithrix* sp. might be lower compared to *Alouatta* sp., as indicated by IHC analysis. However, while the authors compared diagnostic methods, they did not specifically quantify viral load in tissue. Both liver and brain tissues were analyzed using RT-PCR, but it remains unclear whether the elevated Cq values (which typically indicate low viral load) observed in *Callithrix* sp. reflect lower viral loads in both organs or only in the liver. These findings contrast with our antigen detection results, which showed higher YFV antigen levels in the *Callithrix* sp. brain compared to the *Alouatta* sp. (Figure 3B).

It is important to note that NHP species have different distributions across Brazil, and their genetic background—shaped by their evolutionary history and population structure—may influence their susceptibility to YFV. NHPs from the Amazon Basin, an endemic region for YF, are expected to be more resistant to YFV compared to those from YFV-free areas. Additionally, ecological factors unique to each genus/species, such as geographic distribution and behavior patterns, influence infection susceptibility. For instance, the *Callithrix* genus is known to inhabit urban areas [48].

The CNS lesions observed in both genera closely resemble those caused by other flaviviruses, such as West Nile virus (WNV) [49], Dengue virus (DENV) [50], and Zika virus (ZIKV), all of which can lead to neuronal damage. A study by Alcantara et al. (2021) [26], using *Saimiri collins* as an animal model for congenital Zika syndrome, reported moderate CNS lesions, with severe damage observed in the thalamus and cerebellum. The observed alterations included vascular congestion, perivascular edema, inflammatory infiltrates, gliosis, satellitosis, neuronophagia, and necrosis. Although these findings were more pronounced in ZIKV infection, they are similar to those observed in our study (Figure 2).

When YFV antigen localization in the CNS of both genera was analyzed via IHC, a greater positivity was observed in *Alouatta* sp.; however, *Callithrix* sp. exhibited a higher number of YFV antigen-positive cells in the parenchyma and meninges (Figure 3B). On the other hand, this increased antigen detection in the CNS parenchyma of *Callithrix* sp. was not associated with more intense or severe lesions (Figure 3F–H, although the quantification of cell death in the parenchyma of *Callithrix* showed a higher tendency compared to *Alouatta* sp. (Figure 3I–K). An experimental study is needed to better assess the susceptibility of *Callithrix* to YFV infection in the brain.

In the literature, CNS involvement has been described primarily in human cases related to adverse vaccine effects, as reported by Marinho et al. (2019) [22] and a case study by Noronha [51]. Studying the immune response related to both the vaccine and wild virus is crucial to understanding how the immune system acts against YFV infection in the CNS of naturally infected NHPs. This is particularly important given the observed involvement of the CNS in these animals and their significance as sentinels for YFV. Although the CNS is an immunologically privileged organ, limiting damage during the infectious process is critical due to its low regenerative capacity. However, it is important to note that the immune privilege of the CNS is not due to the absence of an immune response but rather its sophisticated regulation [52].

Cytokines modulate immune responses by influencing the differentiation, proliferation, and survival of immune cells, as well as by regulating the production of other cytokines within specific microenvironments. Our analysis of nervous tissue in *Alouatta* sp. and *Callithrix* sp. revealed a complex interplay between pro-inflammatory and anti-inflammatory cytokines, resulting in similar immunopathological profiles across both species (Figure 4 and Figure 5).

The absence of statistically significant differences in cytokine detection suggests that both primate species mount comparable immune responses to YFV infection. Elevated levels of pro-inflammatory cytokines, such as TNF-α and IL-1β, have been associated with significant tissue damage, particularly in the liver, the primary target organ of YFV [49,50,51,52,53]. Conversely, anti-inflammatory cytokines like IL-10 and TGF-β play crucial roles in regulating the immune response and preventing excessive tissue damage. The comparable activation of caspase-3 (Figure 4S–U) in both species further indicates that similar apoptotic pathways are engaged during infection [54,55,56].

These findings underscore the complexity of the immune response in YFV infection and suggest that a balanced regulation between pro-inflammatory and anti-inflammatory cytokines, along with controlled apoptotic activity, is essential for effective infection control and tissue preservation [56,57]. The similar apoptotic responses observed in both species imply that other factors, such as differences in viral tropism, cellular susceptibility, or tissue repair capacity, may contribute to the distinct clinical outcomes observed between *Alouatta* and *Callithrix* species. Understanding these immune response dynamics is crucial for developing targeted therapeutic strategies aimed at modulating the immune response to mitigate tissue damage and improve clinical outcomes in YFV infections.

Although YFV is well-documented for its hepatic tropism, these findings indicate that the CNS could be a secondary target organ depending on the affected NHP species. The results suggest a greater susceptibility of *Callithrix* sp. to YFV neuroinfection compared to *Alouatta* sp., with more extensive viral replication and involvement of the meninges and parenchyma. However, further experimental studies are necessary to analyze other brain compartments and to identify which cell types are most involved in the immune response. Additionally, understanding how different primate species respond to YFV infection could provide insights into the broader impact of the virus on CNS pathology.

## 5. Conclusions

YFV infection in NHPs varies in lethality by species, with *Alouatta* sp. being highly susceptible and experiencing high mortality, while *Callithrix* sp. exhibit variable resistance, with some individuals remaining asymptomatic or developing mild disease [10]. Our findings suggest that these differences in susceptibility and lethality also influence CNS involvement. *Callithrix* sp. showed lower YFV antigen positivity by IHC but exhibited higher YFV replication in CNS tissues, particularly affecting the meninges and parenchyma, as indicated by increased levels of viral antigen labeling. Despite these differences, both genera displayed similar CNS lesion characteristics and immune response profiles, resulting in meningoencephalitis marked by apoptotic cell death and limited inflammatory infiltration. Both species exhibited a balanced immune response, with TNF-α and IL-1β playing key roles in inflammation, while IL-10 and IL-13 were prominent in immunomodulation—patterns that are consistent with other flavivirus infections. Correlation analyses revealed distinct regulatory mechanisms, with *Alouatta* sp. showing a strong association between TGF-β and YFV antigen, whereas *Callithrix* sp. demonstrated significant upregulation of IL-10, suggesting an immunosuppressive role. However, caspase-3 detection was comparable between species, indicating a similar apoptotic response. The greater hepatic involvement observed in *Alouatta* sp. likely contributes to systemic homeostatic imbalance, potentially increasing CNS permeability to YFV infection. In contrast, the higher levels of viral antigen labeling detected in the brains of *Callithrix* sp. suggest increased viral burden in the CNS at the time of sampling with an eventual clearance response. These findings highlight species-specific immune dynamics in YFV pathogenesis and emphasize the critical role of neotropical NHPs as sentinel species.

## Figures and Tables

**Figure 1 viruses-17-00386-f001:**
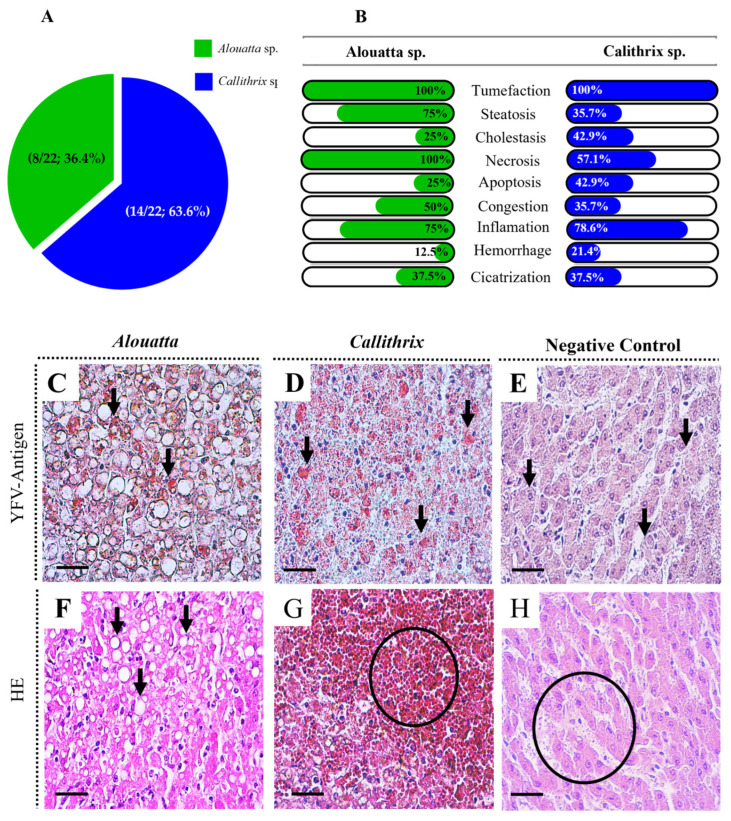
(**A**) Graphic of percentage of number of NHPs per species positive for YFV detection in the liver. (**B**) Frequence of hepatic injuries observed per NHP species. (**C**) Positive immunostaining in hepatocytes of Alouatta (black arrows) (**D**) Positive immunostaining in hepatocytes of Callithrix (black arrows) (**E**) Absence of immunostaining in hepatocytes (black Arrows) (**F**) Steatosis in hepatocytes of Alouatta (black arrows) (**G**) Severe hemorrhage in the liver Callithrix (circle) (**H**) Preservation of hepatocytes and capillary sinusoids in the negative control. Magnification 100× and scale bar 25 μm.

**Figure 2 viruses-17-00386-f002:**
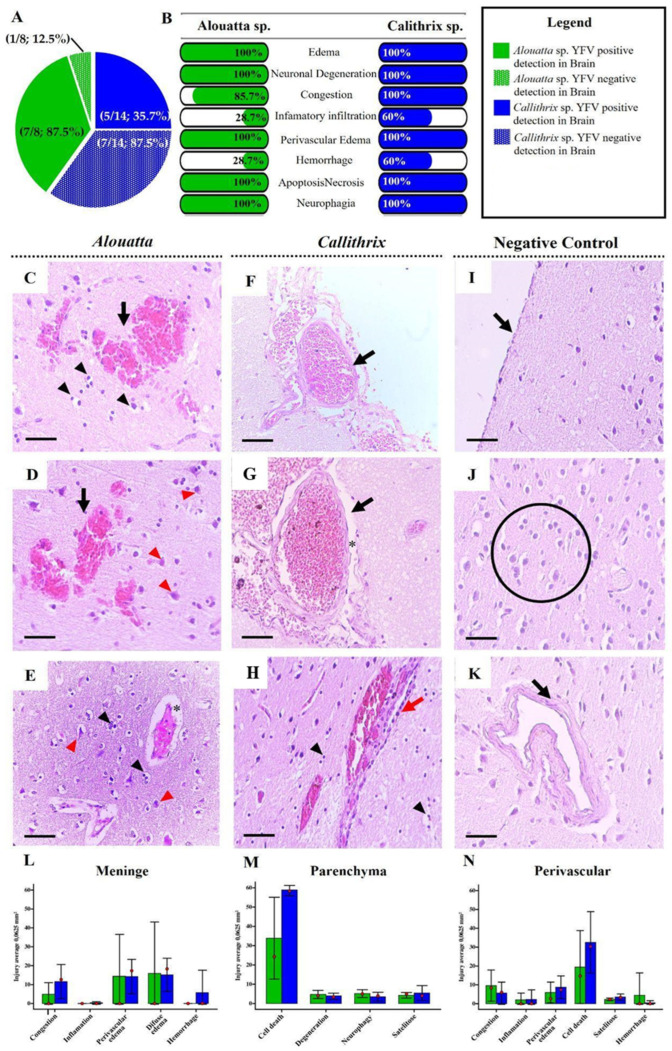
(**A**) Graphic of percentage of number of NHPs per species positive for YFV antigen detection in the brain. (**B**) Frequence of hepatic injuries observed per NHP species. Histopathological analysis shows the main lesions found in the meninges, parenchyma, and perivascular region in the brains of PNHs naturally infected with YFV. (**C**–**E**) Brain of *Alouatta* showing: hemorrhage (black arrow); perivascular edema (*) as well as cells with pyknotic nuclei (black arrowhead) and neuronal degeneration (red arrowhead). (**F**–**H**) Brain of *Callithrix*, showing: (**F**) Vascular congestion (black arrows) in meninge; (**G**) vascular congestion (black arrows) and perivascular edema (*) in the interstitial; (**H**) Inflammatory infiltrate (red arrow) and pyknotic cells around vessels (black arrowhead). (**I**–**K**) Negative control of CNS (**I**) Preservation of the meninges in negative control (black arrow) (**J**) Preservation of the parenchyma in negative control (circle) (**K**) Preservation of the vascular endothelium in negative control (black arrow). All micrographs were obtained at 100× magnification and scale bar 25 µm. (**L**–**N**) Graphs showing the quantification of lesions in the three CNS areas—meninge, parenchyma, and perivascular—stratified by NHP species.

**Figure 3 viruses-17-00386-f003:**
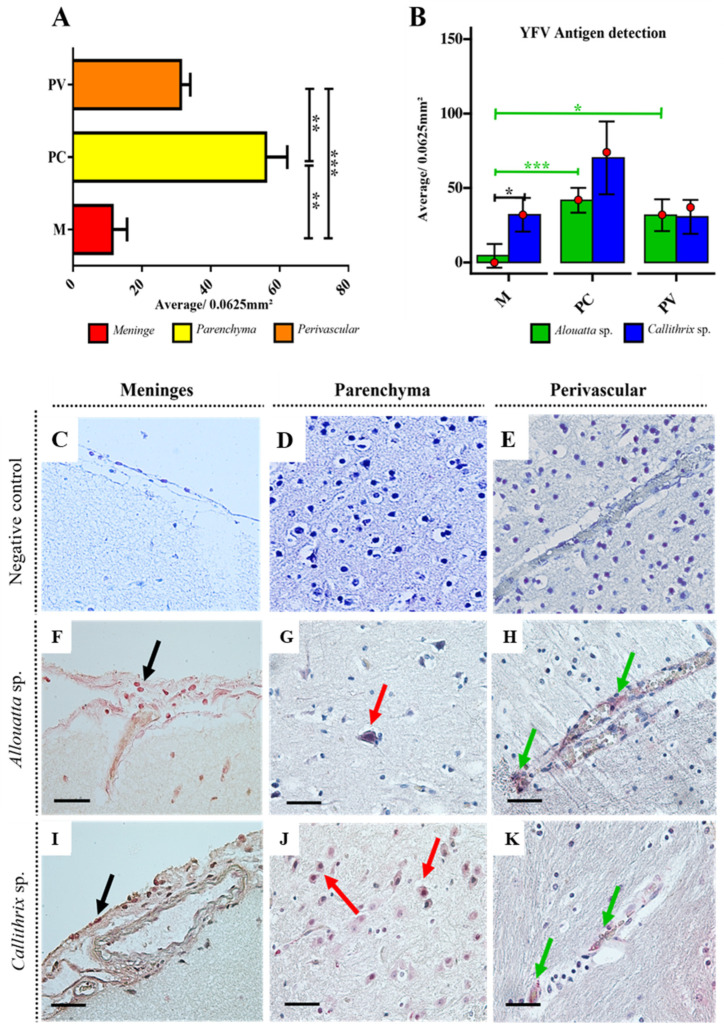
Distribution of YFV antigen in the CNS of neotropical NHP (*Alouatta* sp. and *Callithrix* sp.). (**A**) Graphs showing the total of YFV antigen-positive cells quantification per CNS areas (M = meninges, PC = parenchyma, and PV = perivascular) (**B**) Graphs showing the total of YFV antigen-positive cells quantification stratified by NHP species. The parenchyma contains the highest number of YFV antigen-positive cells in both species. The Krukall–Wallis test indicated a statistically significant difference between the species, especially in M from PC (*p* < 0.0005) and PV (*p* < 0.05). The red dots in the graphs indicate the median. Statistical significance is indicated by asterisks: *p* < 0.05 = Significant (*); *p* < 0.005 = Highly significant (**); *p* < 0.0005 = Extremely significant (***). (**C**–**E**) Micrographs showing the negative control from each CNS area. (**F**) YFV antigen staining in the meninges, in mononuclear cells (black arrow) (**G**) parenchyma in neuron (red arrow), and (**H**) perivascular area in the genus Callithrix with immunomarker for YFV-antigens in endothelial cells (green arrows). (**I**–**K**) YFV antigen staining in the same areas in the genus Alouatta (black arrow for endothelial cells; Red arrows for neurons, and green arrows for endothelial cells). Magnification 100× and scale bar 25 μm.

**Figure 4 viruses-17-00386-f004:**
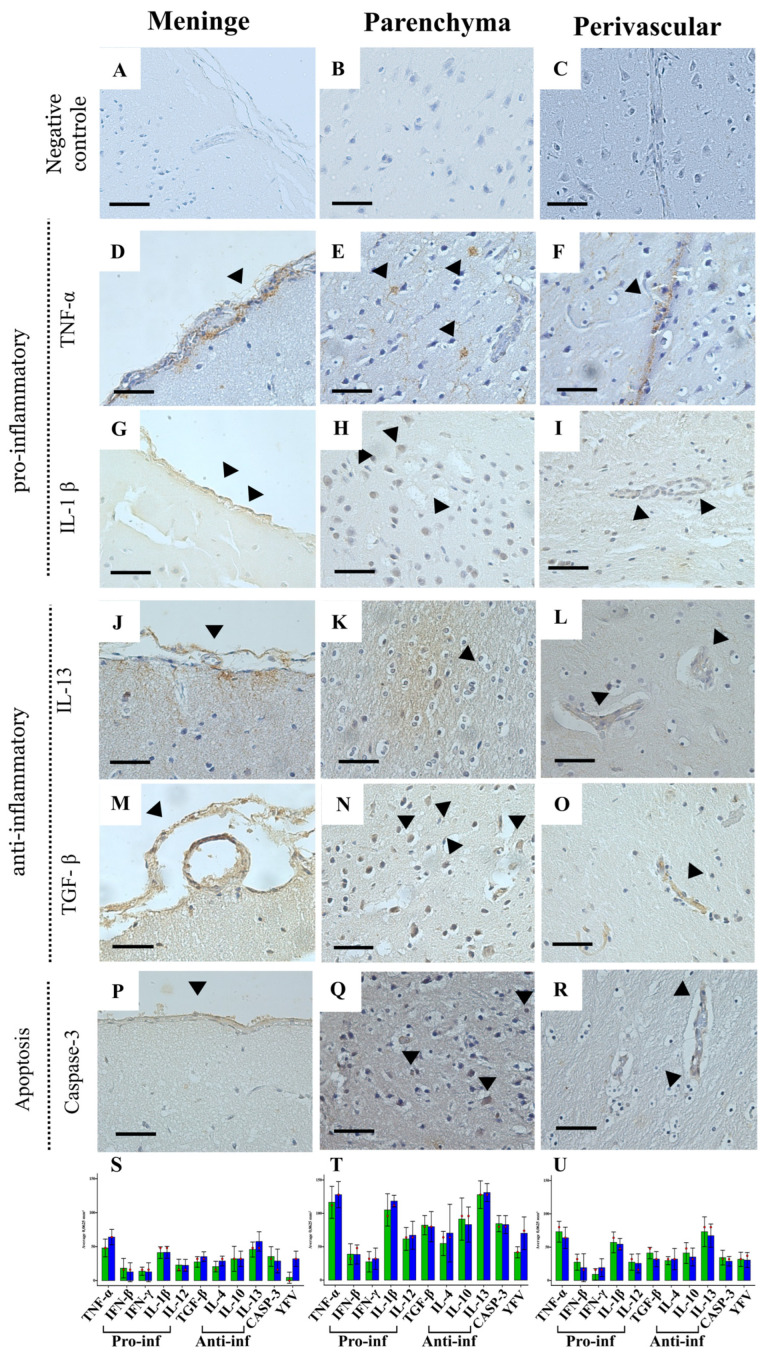
Optical micrographs of immunohistochemistry staining showing the main cytokines highly detected in CNS areas—meninges, parenchyma, and perivascular—of *Callithrix* sp. group. (**A**–**C**) Negative control group; pro-inflammatory cytokines: TNF-α (**D**–**F**) and IL-1β (**G**–**I**); anti-inflammatory cytokines: IL-13 (**J**–**L**) and TGF-β (**M**–**O**); as well as the apoptosis immune marker Caspase-3 (**P**–**R**). Magnification 100× and scale bar 25 µm. The graphs display the quantification of immune response biomarkers in the meninges (**S**), parenchyma (**T**), and perivascular (**U**) areas. **Pro-inf** (Pro-inflammatory) and **Anti-inf** (Anti-inflammatory). Quantification of pro- and anti-inflammatory cytokines showed level variations between *Alouatta* sp. and *Callithrix* sp., but no significant differences (Mann–Whitney test). Notably, all cytokines exhibited significantly higher detection in the parenchyma compared to the meninges and perivascular areas, regardless of the NHP species.

**Figure 5 viruses-17-00386-f005:**
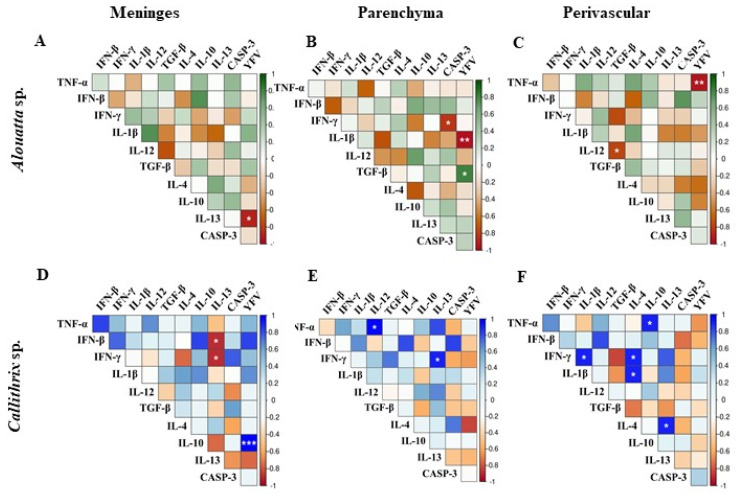
Correlogram showing the correlations among YFV antigen detection, cell death, and the expression of pro- and anti-inflammatory cytokine immunomarkers in the CNS of *Alouatta* sp. and *Callithrix* sp. (**A**–**F**) represent correlation matrices for Alouatta sp. and Callithrix sp. across different CNS regions: (**A**) Meninges, (**B**) Parenchyma, and (**C**) Perivascular regions for Alouatta sp.; (**D**) Meninges, (**E**) Parenchyma, and (**F**) Perivascular regions for Callithrix sp. Correlations were evaluated using Spearman’s rank correlation test. The color scale represents Spearman’s correlation coefficients (rho), ranging from −1 (perfect negative correlation) to 1 (perfect positive correlation), with 0 indicating no correlation. Statistically significant correlations are denoted by white asterisks: *p* < 0.05 = significant (*); *p* < 0.005 = highly significant (**); and *p* < 0.0005 = extremely significant (***).

## Data Availability

Data are contained within the article and Appendix A.

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
