# Peer review of "Histopathological Changes and Immune Response Profile in the Brains of Non-Human Primates Naturally Infected with Yellow Fever Virus"

_viruses, 2025, doi:10.3390/v17030386_

Round 1
Reviewer 1 Report
Comments and Suggestions for Authors
The manuscript ’Histopathological changes and immune response profile in brains of non-human primates naturally infected by yellow fever virus’ by Suzana Ribeiro de Melo Oliveira et al., exanimated histopathological changes in the central nerve systems in two non-human primate species that were infected with yellow fever virus, with focus on three areas, including the meninges, parenchyma, perivascular. Authors later analyzed the correlation between YFV antigens and pro- or anti-inflammatory factors in these areas. The data provided important information not only for YFV infection in the CNS but also for other flavivirus infections.
I have only a few comments.
- In Figure 1, add the actual number in the pia figures, as well as any other information for NHPs, such as sex, age, etc.
- Figure 2, add the information on which cells that YFV antigens were positive.
- In Figure 3, the markings on the figures were shifted, please change, the same with Figure 4, which makes readers hard to understand.
- Figure 5 should be YFV, not anti-YFV. I understand that only co-efficiency was used in Figure 5, in which detailed information has been provided in Supplementary Table 3. If possible, the original data, like in Figures 4 and 2, the data of actual quantification areas, should be added.
Other minor points
- Line 38, 'arthropod-borne' not 'arthropod-born'
- Abbreviation, what is SNC? Should it be CNS?; PNH? Should it be NHP?; IHQ??
Author Response
Question 1: In Figure 1, add the actual number in the pia figures, as well as any other information for NHPs, such as sex, age, etc.
Response: All non-human primate samples were obtained from epidemiological investigations conducted in different Brazilian states. The information provided on the forms was incomplete, and it was not possible to retrieve data on age, sex, or other relevant information for the study.
Question 2: Figure 2, add the information on which cells that YFV antigens were positive. Response: included
Question 3: In Figure 3, the markings on the figures were shifted, please change, the same with Figure 4, which makes readers hard to understand. Response: The figures were replaced, improving the image quality.
Question 4: Figure 5 should be YFV, not anti-YFV. I understand that only co-efficiency was used in Figure 5, in which detailed information has been provided in Supplementary Table 3. If possible, the original data, like in Figures 4 and 2, the data of actual quantification areas, should be added. Response: We have included tables with raw data corresponding to Figures 2, 3, 4, and 5.
Other minor points
- Line 38, 'arthropod-borne' not 'arthropod-born' change made
- Abbreviation, what is SNC? Should it be CNS?; PNH? Should it be NHP?; IHQ?? We updated all abbreviations to the English format.

Reviewer 2 Report
Comments and Suggestions for Authors
In this work, Oliveira et al. investigate characteristics of YF disease in two NHP populations that are susceptible to sylvatic circulation of YFV – the commonly infected Alouatta sp. and the less-characterized Callithrix sp. The authors utilize liver and brain tissue from YFV-infected animals and utilize staining and microscopy to identify pathological features and immune responses in the brains of animals. From these samples, the authors were able to identify brain pathology from YFV infection occurring in both groups of animals, but noted different immune responses. The results highlight the need to investigate other organs besides the liver affected in YFV-infected wild animals, and also reveal differences in the tissue tropism and responses to YFV between the two NHP genera.
The manuscript is interesting and investigates a very understudied subject in YFV virology and monitoring. However, the writing is often unclear and the authors need to include more images / data as well as mock-infected or healthy tissue to support their quantifications and claims of pathology.
Major Concerns
· All axis labels for bar graphs (2A, 2K-2M, 3C, 3F, 3I, 4D, 4H, 4I-4M, 4Q, 4U, 4Y) are very small and should be made larger.
· The authors have scale bars on their images that don’t have measurements associated.
· Figure 1 – The authors should include some representative images of the liver pathology features they discuss.
· In Figure 3 and possibly Figure 4, many of the arrows and symbols have moved and need to be re-positioned.
· In Figures 3 and 4, the authors should include images from mock-infected or animals with healthy brains for the reader to compare the diseased pictures against.
· In Figures 3 and 4, the authors should specify which species the images are from.
· For the bar graphs in Figure 4, do the authors have quantifications from mock-infected or healthy tissue to help the reader understand the immune response happening to YFV?
· In regard to Figure 5, the authors should include some interpretations of what the correlations between YFV antigen or between cytokines mean. These could be added to the Results or Discussion section.
· In Figures 4 and 5, are the authors using antibodies against cleaved caspase-3 or total caspase-3?
· Lines 218-229 and 243-248: The authors should refer to specific images /data for each feature they list instead of just listing the entire figure at the end of the paragraph.
· Section 3.2, lines 242-267: The authors should introduce the immune profiles / cytokines they are exploring more in-depth. They should give some brief background on the cytokines, Th1 vs. Th2 responses, and why these are relevant here.
· In the text for Figures 4 and 5 the authors refer to IFN-α but show no data for it. Is IFN-α correct or should this actually be IFN-γ?
· Lines 244-245: What are the authors comparing TGF-β and IFN to? It’s unclear how they’re concluding they are either lower or higher.
· In Figure 4, the authors should include images in the Supplementary Data for the cytokines graphed but not pictured (IFN, IL-12, IL-4, etc.). They should also include mock images and quantifications for comparison.
· Lines 258-267: The authors should include some interpretations of what these correlations mean, either here or in the Discussion.
· Lines 363-367: Where are the authors getting the basis for these claims? How can they conclude IFN-y and IFN-b were not significantly expressed to contain viral replication?
· Lines 396-400: The authors should clarify this caveat in the Results section and data. How many animals were analyzed / used for correlation analysis? How are the statistics affected? It is unclear exactly what the authors did and the caveats that should be taken into consideration.
Minor Concerns/language edits
· Lines 40, 46, 53, 55, 92, 217, 233, 424 – PNH should be NHP
· Lines 244, 245, Figure 4, Figure 5, line 382 - INF should be IFN
· Lines 126, 128, 149, 152, 154 – T.A. and A.T. should be R.T.
· Lines 212, 217, 241, 242 – SNC should be CNS
· Lines 211, 250 – IHQ should be IHC
· Line 44 – reported, not reporting
· Line 51: Main, not mainly
· Line 86: genus specific what?
· Line 88: This should be genera, not genus
· Lines 85-89: These sentences are confusing grammatically
· Line 95: Did the authors really mean to use the word “casuistry?”
· Line 115 – The authors should include materials and methods for their H&E staining protocol rather than just mentioning it.
· Line 138 – Were slides analyzed by a trained veterinary pathologist? Is the “doctor” a human physician or a veterinarian?
· Line 151-152 – Is 10% Tween-20 correct?
· Lines 188-198: The authors should include the number of monkeys for each percentage as they do in the next paragraph (i.e. 1/8; 12.50%)
· Figure 1 - The authors should include the number of monkeys as well as percentages for each feature.
· Line 200: Percentage. The authors used the Portuguese word
· Figure 2A – x-axis needs a label
· Figures 2K, L, M – What does “YFV antigen average” mean? The average number of positive cells?
· Figure 2 Legend: The authors should include a description of what the arrows in E-J are pointing out. They should also mention the statistical test(s) used and what p-values the stars represent.
· Line 240 – What does “(seta)” mean?
· Line 240 – pyknotic, not picnotic
· Y-axes for 3C, 3F, 3I: Injury, not “injure.” Also, what does this mean? Is this total number of cells displaying these injuries?
· Some of the features described in Figure 3D are unclear. Do the authors have any other images of these features they could include in the supplement?
· For Figure 3, 3re there any literature references for these disease features in humans or NHPs?
· Line 250 – main, not “mainly”
· In Figure 4, 4V, 4X, 4W should say Apoptosis not Apoptose
· In Figure 4, the y-axis labels are unclear as to what “average” means
· It could be helpful for the authors to include supplemental images from whatever monkey is not shown in the main figure.
· Line 252: Marker, not “mark”
· Line 254-255: I think this should be “comparing Callithrix sp. To Alouattas sp.”
· Line 256: Extra () added
· Line 256: Regardless, not regarding
· Line 258 – More introduction to doing correlograms should be included.
· In Figure 5, does this correlation take into account of many or how few animals were in some categories? How many animals / images were used for each?
· Line 272: [B]razil
· Lines 313-315: What are these percentages referring to? YFV positive? Disease markers positive?
· Line 321 – Genetic history? What does this mean?
· Lines 333-336: Refer back to a figure / data
· Line 355&357: I think genders should be “genera”
· Lines 348-357: Refer to Figure / data
· Lines 353-357: What do the authors mean by “markings?”
· Lines 371-375: I don’t really understand what is trying to be said here
· Lines 376-381: This part is a little unclear. The authors should refer to their figures / data and clarify what they think is happening based on relevant literature
· Lines 382-386: IFN-a and IFN-b are classic antiviral signaling molecules and should be expected to be induced during flavivirus infection. The authors should refer to their data and clarify the point they are trying to make here.
· Lines 394-396: This sentence should be rewritten
· Referring to line 411: “[infection] is often lethal.” Do the authors know that infection is often lethal? It is unclear how often these animals get infected and die since the total number of samples was lower.
· Table S1:
o Biomarkers
o Animal Origen – Rabbit, not Habbit
o Is IFN-b actually anti-inflammatory?
o Apoptosis, not apoptose
· Table S2:
o ID numbers are shifted
o Species, not specie
· Table S3:
o P-value, not p-valor
o Moderate significant, not “modarate.”
o What does the b refer to?
o Is this actually INF-a or should it be IFN-y?
· Lines 424-425: This sentence is in Portuguese
Author Response
Question 1: All axis labels for bar graphs (2A, 2K-2M, 3C, 3F, 3I, 4D, 4H, 4I-4M, 4Q, 4U, 4Y) are very small and should be made larger.The authors have scale bars on their images that don’t have measurements associated. In Figure 3 and possibly Figure 4, many of the arrows and symbols have moved and need to be re-positioned.
Response: The figures were replaced considering your observation.
Question 2: Figure 1 – The authors should include some representative images of the liver pathology features they discuss.
Response: include in figure 1
Question 3: In Figures 3 and 4, the authors should include images from mock-infected or animals with healthy brains for the reader to compare the diseased pictures against.
Response: we have included an image of a non-infected animal brain as a mock in figure 2 and 3.
Question 4: In Figures 3 and 4, the authors should specify which species the images are from.
Response:The images of hepatic and CNS lesions in Figures 3 and 4 (update as figure 1 and 2) are representative of both species. We selected the best images regardless of species."
Question 5: For the bar graphs in Figure 4, do the authors have quantifications from mock-infected or healthy tissue to help the reader understand the immune response happening to YFV?
Response: Yes, we evaluated the immune response in animals that were not infected with the Yellow Fever virus, considering them as mock controls (n=2). however they show less than 3 immunolabeling per field. We used their data as a threshold for cytokine detection to identify relevant markings.
Question 6: In regard to Figure 5, the authors should include some interpretations of what the correlations between YFV antigen or between cytokines mean. These could be added to the Results or Discussion section.
Response: The correlation between elevated IL-10 levels and the presence of the yellow fever virus (YFV) antigen suggests an adaptive immune strategy to balance the response to infection. The increased production of IL-10 may represent a mechanism by which the body reduces viral pathogenicity and minimizes tissue damage, promoting a less destructive immune response. Thus, in contexts where the viral antigen is present in high concentrations, IL-10 elevation may contribute to mitigating the adverse effects of infection and modulating inflammation throughout the pathogenesis of yellow fever.
Question 7: In Figures 4 and 5, are the authors using antibodies against cleaved caspase-3 or total caspase-3?
Response: total caspase-3
Question 8: Lines 218-229 and 243-248: The authors should refer to specific images /data for each feature they list instead of just listing the entire figure at the end of the paragraph.
Response: information included
- Question 9: Section 3.2, lines 242-267: The authors should introduce the immune profiles / cytokines they are exploring more in-depth. They should give some brief background on the cytokines, Th1vs. Th2 responses, and why these are relevant here.field
Response: We have incorporated additional information in the methodology section to provide a more in-depth introduction to the immune profiles and cytokines analyzed. This includes a brief background on the Th1 vs. Th2 responses and their relevance to understanding the tissue response induced by YFV infection in Alouatta and Callithrix.
- Question 10: In the text for Figures 4 and 5 the authors refer to IFN-α but show no data for it. Is IFN-α correct or should this actually be IFN-γ?
Response: change made
- Question 11: Lines 244-245: What are the authors comparing TGF-β and IFN to? It’s unclear how they’re concluding they are either lower or higher.
Response: change made
- Question 12:In Figure 4, the authors should include images in the Supplementary Data for the cytokines graphed but not pictured (IFN, IL-12, IL-4, etc.). They should also include mock images and quantifications for comparison.
Response: We included figures (including images of the negative controls) as well as tables with the raw data from the graphs in the supplementary material. The quantification data of cytokines based on the negative control group (n=3) were not included as they showed fewer than 3 markings per microscopic field. Instead, they were used as the threshold in the graphs (X axis).
- Question 13: Lines 258-267: The authors should include some interpretations of what these correlations mean, either here or in the Discussion.
Response:The correlation between elevated IL-10 levels and the presence of the yellow fever virus (YFV) antigen suggests an adaptive immune strategy to balance the response to infection. The increased production of IL-10 may represent a mechanism by which the body reduces viral pathogenicity and minimizes tissue damage, promoting a less destructive immune response. Thus, in contexts where the viral antigen is present in high concentrations, IL-10 elevation may contribute to mitigating the adverse effects of infection and modulating inflammation throughout the pathogenesis of yellow fever.
- Question 14: Lines 363-367: Where are the authors getting the basis for these claims? How can they conclude IFN-y and IFN-b were not significantly expressed to contain viral replication?
Response: we review all data about cytokine.
- Question 15: Lines 396-400: The authors should clarify this caveat in the Results section and data. How many animals were analyzed / used for correlation analysis? How are the statistics affected? It is unclear exactly what the authors did and the caveats that should be taken into consideration.
Response: The limited number of animals and samples analyzed represents a limitation of the study and may influence the statistical robustness of the findings. To minimize these effects, as previously mentioned, we used Spearman's correlation test, which is more robust for small sample sizes since it does not assume a normal data distribution and is less influenced by extreme values. Nevertheless, we emphasize that the interpretation of the results should be approached with caution, considering the inherent sample limitations of the study.
Minor Concerns/language edits
- Lines 40, 46, 53, 55, 92, 217, 233, 424 – PNH should be NHP change made
- Lines 244, 245, Figure 4, Figure 5, line 382 - INF should be IFN change made
- Lines 126, 128, 149, 152, 154 – T.A. and A.T. should be R.T.change made
- Lines 212, 217, 241, 242 – SNC should be CNS change made
- Lines 211, 250 – IHQ should be IHC change made
- Line 44 – reported, not reporting change made
- Line 51: Main, not mainly change made
- Line 86: genus specific what? corrected to "NHP species"
- Line 88: This should be genera, not genus change made
- Lines 85-89: These sentences are confusing grammatically rewritten text
- Line 95: Did the authors really mean to use the word “casuistry?” corrected to "NHP samples"
- Line 115 – The authors should include materials and methods for their H&E staining protocol rather than just mentioning it. included
- Line 138 – Were slides analyzed by a trained veterinary pathologist? Is the “doctor” a human physician or a veterinarian?
Response: Yes, the slides were analyzed by both a trained veterinary pathologist and a physician pathologist. Our team includes a veterinary pathologist (B.N.A.) and two physician pathologists (E.R.M.C. and M.T.F.A.). Dr. M.T.F.A. is a professor with extensive experience studying YFV in both humans and animal models.
- Line 151-152 – Is 10% Tween-20 correct? change made
- Lines 188-198: The authors should include the number of monkeys for each percentage as they do in the next paragraph (i.e. 1/8; 12.50%) change made
- Figure 1 - The authors should include the number of monkeys as well as percentages for each feature. included
- Line 200: Percentage. The authors used the Portuguese word change made
- Figure 2A –X-axis needs a label included
- Figures 2K, L, M – What does “YFV antigen average” mean? The average number of positive cells?
Response: Yes, "YFV antigen average" refers to the average number of cells positive for yellow fever virus (YFV) antigen, as detected by immunohistochemistry (IHC). We have revised the text to clarify this point and ensure better understanding.
- Figure 2 Legend: The authors should include a description of what the arrows in E-J are pointing out. They should also mention the statistical test(s) used and what p-values the stars represent.
Response: We have updated the figure legend to include a description of what the arrows in panels E-J are pointing out. Additionally, due to the use of multiple statistical tests in our analyses, the absolute p-values and the specific statistical tests applied have been detailed in the main text for clarity and to avoid redundancy in the figure legend.
- Line 240 – What does “(seta)” mean? included in the legend
- Line 240 – pyknotic, not picnotic change made
- Y-axes for 3C, 3F, 3I: Injury, not “injure.” Also, what does this mean? Is this total number of cells displaying these injuries?
Response: change made
- Some of the features described in Figure 3D are unclear. Do the authors have any other images of these features they could include in the supplement?
Response: we replace the figures
- For Figure 3, 3re there any literature references for these disease features in humans or NHPs?
Response:To date, there are few reports in the literature regarding CNS lesions caused by YFV in non-human primates (NHPs), with most studies being experimental in nature. Notably, we identified Cunha et al. (2019) [8], which provides an analysis of diagnostic methods in neotropical NHPs. The majority of existing literature, however, focuses exclusively on CNS lesions in humans.
- Line 250 – main, not “mainly” change made
- In Figure 4, 4V, 4X, 4W should say Apoptosis not Apoptose change made
- In Figure 4, the y-axis labels are unclear as to what “average” means. included
- It could be helpful for the authors to include supplemental images from whatever monkey is not shown in the main figure. included supplementary figure 1
- Line 252: Marker, not “mark” change made
- Line 254-255: I think this should be “comparing Callithrix sp. To Alouattas sp.”change made
- Line 256: Extra () added extra () excluded
- Line 256: Regardless, not regarding change made
- Line 258 – More introduction to doing correlograms should be included.
- In Figure 5, does this correlation take into account of many or how few animals were in some categories? How many animals / images were used for each?
Response: A total of 22 brain samples from primates naturally infected with YFV were analyzed, including 8 from Alouatta sp. and 14 from Callithrix sp. Correlation analyses were conducted separately for each taxonomic group and for different anatomical categories (parenchyma, meninges, and perivascular region). However, the number of cytokine quantification observations varied between groups: 5 observations were used for Alouatta sp., while 7 observations were considered for Callithrix sp.
Due to the small sample size, Spearman’s correlation test was chosen, as it is a non-parametric approach more suitable for small datasets, given that it does not assume normality in variable distribution and is less sensitive to extreme values. Thus, the correlations presented reflect the interactions between the analyzed biomarkers within each group and anatomical category, considering the study’s sample size limitations.
Each image represents a correlogram corresponding to a specific anatomical category (e.g., parenchyma). Within each taxonomic group (Alouatta and Callithrix) and for each anatomical category, the Spearman correlation coefficient was calculated for all biomarker combinations.
- Line 272: [B]razil change made
- Lines 313-315: What are these percentages referring to? YFV positive? Disease markers positive?
Response: This percentage refers to the number of individuals with a confirmed diagnosis of yellow fever in liver samples who also had yellow fever antigen detected in the brain.
- Line 321 – Genetic history? What does this mean?
Response: By "genetic history," we are referring to the genetic characteristics and evolutionary differences that distinguish various Neotropical primate species. the text has been changed for greater understanding
- Lines 333-336: Refer back to a figure / data change made
- Line 355&357: I think genders should be “genera” change made
- Lines 348-357: Refer to Figure / data included
- Lines 353-357: What do the authors mean by “markings?”
Response: The term "markings" in this context refers to immunolabeling or Immunomarkers, indicating areas where specific cytokines were detected using immunohistochemistry (IHC). We replaced "markings" in the text with "Immunomarkers."
- Lines 371-375: I don’t really understand what is trying to be said here
Response: We have modified the text. I hope it is now clearer.
- Lines 376-381: This part is a little unclear. The authors should refer to their figures / data and clarify what they think is happening based on relevant literature
Response: The text was modified, being including your suggestions - Lines 382-386: IFN-a and IFN-b are classic antiviral signaling molecules and should be expected to be induced during flavivirus infection. The authors should refer to their data and clarify the point they are trying to make here. Response: in flaviviruses IFN-a and IFN-b is higher in serum, but in tissue can variety. included
- Lines 394-396: This sentence should be rewritten
Response: The text was modified, being including your suggestions
- Referring to line 411: “[infection] is often lethal.” Do the authors know that infection is often lethal? It is unclear how often these animals get infected and die since the total number of samples was lower.
Response: YFV infection in non-human primates (NHPs) exhibits varying degrees of lethality depending on the species. In the Americas, certain species, particularly Alouatta sp are highly susceptible to YFV and often experience high mortality rates during outbreaks. Conversely, other species, such Callithrix sp, exhibit variable resistance to YFV infection, with some individuals remaining asymptomatic or experiencing milder disease (see Ref 10 https://doi.org/10.1590/0037-8682-0363-2015). - Table S1:
o Biomarkers change made
o Animal Origen – Rabbit, not Habbit change made
o Is IFN-b actually anti-inflammatory?
Response: IFN-β is predominantly considered anti-inflammatory due to its ability to downregulate the production of pro-inflammatory cytokines such as TNF-α and IL-1β while promoting the expression of IL-10, an anti-inflammatory cytokine. Additionally, IFN-β modulates immune responses by reducing excessive immune activation, limiting leukocyte migration, and suppressing inflammatory signaling pathways. Although it may have context-dependent immunostimulatory effects, its overall role in immune regulation supports its classification as an anti-inflammatory cytokine, particularly in the context of neuroinflammation and autoimmune diseases like multiple sclerosis.
o Apoptosis, not apoptose change made
- Table S2:
o ID numbers are shifted change made
o Species, not specie change made
- Table S3:
o P-value, not p-valor change made
o Moderate significant, not “modarate.” change made
o What does the b refer to? “b” doesn´t exist. it is change to “p”
o Is this actually INF-a or should it be IFN-y? change made
- Lines 424-425: This sentence is in Portuguese change made

Reviewer 3 Report
Comments and Suggestions for Authors
Major comments
1. Section on Materials and Methods of Immunohistochemistry for immune response and Quantitative analysis and photodocumentation (Line 169~ 172): It is need 10 primary immune-related antibodies to detect 10 section of brain tissue each of Alouatta (7/8) and Calithrix (5/14), is it performed Individually in 7 Alouatta and 5 Calithrix brains?
2. Is the Immunomarkers quantified by randomly selecting 10 high-power fields for the meningeal, perivascular and neural parenchyma regions of 7 Alouatta and 5 Callithrix brains?
Minor comments:
1. Line 25: “ perivascular edema” is duplicated
2. Line 40, 46, 53, 55, 92, 217, 233: “PNH” might be corrected to “NHPs”
3. Line 75, 489: the ref [16] is not related.
4. Line 85, 508: the ref [25] is not related.
5. Line 92: “humans and” might be deleted.
6. Line 93: “during 2017” may be ”during 2015-2017”
7. Line 126, 128, 149, 152, 154: “ A.T.” might be “R.T.”
8. line 197, Figure 1-B: “apoptosis was more evident (50%)”, but in Figure 1-B, the apoptosis of Aloutta sp. showed only 25%.
9. Line206: “neuronal degeneration” might be “axonal degeneration”.
10. Line 209: “and meninges” might be “perivascular”.
11. Line 211, 250: “IHQ” might be “IHC”.
12. Line 211, 212, 217, 141, 142: “SNC” might be “CNS”.
13. Line 232-241: the order of Figure 3 legend is not correct.
14. Line 244-245: “INF-α and INF-β” might be “ TNF-α and INF-g”
15. Line 254: “D, H, K, I, J, K,…” --“ K” is duplicated.
16. Line 254: ” Alouatta” is redundancy.
17.Line 286: “most” might be “Most”.
18. Line 304: “ hamsters [34]” may be incorrect.
19. Line 319: [36] showed that howler (Alouatta) monkey, but not Callithrix, is sensitive to YFV.
20. Line 327: “ [41]” may be incorrect.
21. Line 349,382: “IFN-α” might be “INF-g”
22. Line 355, 357: “genders” might be “genus”
23. Line 386: “ [43]” might be “ [45]”
24. Line 389: the ref [52] is not related.
Author Response
Question 1: Section on Materials and Methods of Immunohistochemistry for immune response and Quantitative analysis and photodocumentation. (Line 169~ 172): It is need 10 primary immune-related antibodies to detect 10 section of brain tissue each of Alouatta (7/8) and Calithrix (5/14), is it performed Individually in 7 Alouatta and 5 Calithrix brains?
Question 2: Is the Immunomarkers quantified by randomly selecting 10 high-power fields for the meningeal, perivascular and neural parenchyma regions of 7 Alouatta and 5 Callithrix brains?
Response: 7 Alouatta and 5 Callithrix brains were evaluated. Each brain was assessed in three areas: meninges, parenchyma, and perivascular regions, with 3 fields examined per area. The text has been modified for better understanding.
Minor comments:
- Line 25: “ perivascular edema” is duplicated - change made
- Line 40, 46, 53, 55, 92, 217, 233: “PNH” might be corrected to “NHPs” - change made
- Line 75, 489: the ref [16] is not related. - change made
- Line 85, 508: the ref [25] is not related. - change made
- Line 92: “humans and” might be deleted. Yes. but we focus only in monkeys
- Line 93: “during 2017” may be ”during 2015-2017”change made
- Line 126, 128, 149, 152, 154: “ A.T.” might be “R.T.” change made
- line 197, Figure 1-B: “apoptosis was more evident (50%)”, but in Figure 1-B, the apoptosis ofAlouttasp. showed only 25%.change made
- Line206: “neuronal degeneration” might be “axonal degeneration”. change made
- Line 209: “and meninges” might be “perivascular”. No, the perivascular area refers to the vessels located within the parenchyma. When we discuss inflammation in the parenchyma, we are referring to inflammatory infiltrates in the interstitial space.
- Line 211, 250: “IHQ” might be “IHC”.change made
- Line 211, 212, 217, 141, 142: “SNC” might be “CNS”. change made
- Line 232-241: the order of Figure 3 legend is not correct. change made
- Line 244-245: “INF-α and INF-β” might be “ TNF-α and INF-g” change made
- Line 254: “D, H, K, I, J, K,…” --“ K” is duplicated. change made
- Line 254: ” Alouatta” is redundancy. change made
17.Line 286: “most” might be “Most”. change made
- Line 304: “ hamsters [34]” change made
- Line 319: [36] showed that howler (Alouatta) monkey, but not Callithrix, is sensitive to YFV.
- Line 327: “ [41]” change made
- Line 349,382: “IFN-α” might be “INF-g” change made
- Line 355, 357: “genders” might be “genus”change made
- Line 386: “ [43]” might be “ [45]” - change made
- Line 389: the ref [52] is not related. - change made

Reviewer 4 Report
Comments and Suggestions for Authors
The text is well written and shows the importance of obtaining knowledge of other areas affected by YFV in addition to the organs already known. This information is important for surveillance studies of sentinel animals as well as for preserving the species.
#minor
L40: review the acronym PNH throughout the text and replace it by NHP
L107: what were the conditions of the carcasses at the time of sample collection? What were the conditions of death of these animals? Time of death, etc. Describe the history of sample collection and storage conditions
L117: There is no Chart 1 in the manuscript
L119: “using polyclonal antibodies specific for YFV,…” has this antibody been validated? Is there any reference?
L129: SAAP solution: specify which solution it is. Commercial? If it is in-house, describe the composition
Supplementary Table 2: title is not in English
L232: Figure 3: review the arrows and signs because the image was distorted
Author Response
L40: review the acronym PNH throughout the text and replace it by NHP change made
L107: what were the conditions of the carcasses at the time of sample collection? What were the conditions of death of these animals? Time of death, etc. Describe the history of sample collection and storage conditions
Response: All samples in this study were sent to the Evandro Chagas Institute (IEC) by various surveillance services from different Brazilian states. Unfortunately, the epidemiological records lack detailed information regarding the animals' deaths; many were found deceased in the wild. All samples were preserved in 10% buffered formalin for transport. Histopathological analyses were conducted in the Department of Pathology at IEC, with confirmation of infection by the yellow fever virus initially in the liver. Our study focuses on CNS analysis.
L117: There is no Chart 1 in the manuscript : change to Supplementary Table 1
L119: “using polyclonal antibodies specific for YFV,…” has this antibody been validated? Is there any reference? This polyclonal antibodies
Response: The polyclonal anti-YFV antibodies used in this study have been routinely employed in the diagnosis of yellow fever by the Department of Pathology at IEC, which is a national reference alongside the Ministry of Health. These antibodies have been validated in previous studies, such as in Ribeiro, Y.P.; Falcão, L.F.M.; Smith, V.C.; de Sousa, J.R.; Pagliari, C.; Franco, E.C.S.; Cruz, A.C.R.; Chiang, J.O.; Martins, L.C.; Nunes, J.A.L.; et al. Comparative Analysis of Human Hepatic Lesions in Dengue, Yellow Fever, and Chikungunya: Revisiting Histopathological Changes in the Light of Modern Knowledge of Cell Pathology. Pathogens 2023, 12, 680. https://doi.org/10.3390/pathogens12050680
L129: SAAP solution: specify which solution it is. Commercial? If it is in-house, describe the composition. text modified
Supplementary Table 2: title is not in English change made
L232: Figure 3: review the arrows and signs because the image was distorted change made

Round 2
Reviewer 2 Report
Comments and Suggestions for Authors
The authors have made extensive changes to the manuscript which has greatly improved it. The data and writing are much clearer, and all of the author’s claims are backed by specific figures. There are a few small concerns to be addressed, but afterwards this work is acceptable. There are also grammatical issues, but these should be picked up in the editing and proofing process.
Major Concerns:
- Lines 172-179 – This is a great explanation, but it is strange that it is in the materials and methods section. I would suggest moving it to the Results section.
- Line 253-272 – I think these paragraphs actually refer to Figure 1 and are mislabeled.
- Line 277-278 – is it 12 or 13 monkeys? It is different in the parentheses.
- Figure 4 – Double check that the arrows in 4D-4R are placed properly
- Lines 457-465 – I’m not sure that the authors can conclude anything about persistent infection here. They should be very careful with the wording of their conclusions as they don’t have multiple timepoints.
- Line 681 – Higher viral RNA? Is there a reference for this? The authors never show higher viral RNA levels in their samples
- Line 696 – Add reference
- Lines 703-704 – Again, what viral RNA are the authors referring to? Also, they cannot make any conclusions about viral persistence from these data.
Minor Concerns
- Line 4 – remove INF
- Line 21 – this sentence sounds good but needs to be rearranged
- Line 25 – tendency of is the wrong phrase
- Line 27-28 – Limited inflammatory infiltration
- Line 42 – remove “the” YFV
- Line 51 – from, not during
- Line 69 – is “a” NHP model
- Line 69 – “being” instead of “are”
- Line 122 – superscript mm3
- Line 136 – “submitted to the IHC technic” sounds strange
- Line 150 – 1:500?
- Line 184 – subscript H2O2
- Line 208 – superscript mm2
- Line 242 – This should actually be “We selected”
- Line 409 – reference for “linked to the Th2 response?”
- Line 411 – Regarding “caspase-3” as an apoptosis marker.
- Figure 5 – TNF-a in 5E is slightly cut off and the graph in 5A is cut off.
- Figure 5 legend – Include statistics description and what the values on the bar from -1 to 1 are.
- Line 504 – Italicize Alouatta
- Line 565/566 – Italicize Callithrix and Alouatta
- Line 640 – Italicize
- Supplementary table 1 – Dropped an “s” from species
- Supplementary Figure 1 – Some labeling is not in English
- Supplementary Table 6 – Typo: Interticial edema
- Supplementary Table 7 – Typo: Satelitose
- Supplementary Table 8 – Line numbers obscure table
- Supplementary Table 8 – Typo: Satelitose
- Supplementary Table 9 – Check column titles, i.e. P.ajustado
Comments on the Quality of English Language
There are still some grammatical issues, but these should be picked up in the editing and proofing process
Author Response
Major Concerns:
- Lines 172-179 – This is a great explanation, but it is strange that it is in the materials and methods section. I would suggest moving it to the Results section. R: We appreciate the observation made, we have included this information in the methodology section to justify the choice of cytokines related to the study.
- Line 253-272 – I think these paragraphs actually refer to Figure 1 and are mislabeled. R: Change made.
- Line 277-278 – is it 12 or 13 monkeys? It is different in the parentheses. R: There were 12 (Change made).
- Figure 4 – Double check that the arrows in 4D-4R are placed properly. R: Change made.
- Lines 457-465 – I’m not sure that the authors can conclude anything about persistent infection here. They should be very careful with the wording of their conclusions as they don’t have multiple timepoints. R: Thank you very much for your consideration. In the text range indicated by the reviewer, we present observations obtained when comparing two populations of hosts analyzed. Although our findings demonstrated greater persistence of the viral antigen in the Callithrix population, we emphasize the need for ancillary and subsequent studies that can better assess the involvement of the CNS in this group. When new experimental procedures are performed, we will certainly analyze these aspects and report them in future studies.
- Line 681 – Higher viral RNA? Is there a reference for this? The authors never show higher viral RNA levels in their samples. R: Thank you very much for your valuable observations. "Viral persistence" has been replaced with "In contrast, the higher levels of viral antigen labeling detected in the brains" making it clear that the data reflects a single time point without suggesting persistence.
- Line 696 – Add reference. R: We appreciate your contribution to our work, but we were unable to identify the information in our text to make the suggested changes, as the aforementioned line is a reference.
- Lines 703-704 – Again, what viral RNA are the authors referring to? Also, they cannot make any conclusions about viral persistence from these data. R: Thank you very much for your valuable observations, The text was changed as it involves viral antigen marking “as indicated by increased levels of viral antigen labeling”.
Minor Concerns
- Line 4 – remove INF. R: Change made.
- Line 21 – this sentence sounds good but needs to be rearranged. R: Thank you for your observation, but we were unable to identify the information in the indicated line to make the suggested changes.
- Line 25 – tendency of is the wrong frase. R: Change made.
- Line 27-28 – Limited inflammatory infiltration. R: Change made.
- Line 42 – remove “the” YFV. R: Change made.
- Line 51 – from, not during. R: Change made.
- Line 69 – is “a” NHP model. R: Change made.
- Line 69 – “being” instead of “are”. R: Change made.
- Line 122 – superscript mm3 . R: Change made.
- Line 136 – “submitted to the IHC technic” sounds strange. R: Change made.
- Line 150 – 1:500? R: Thank you for your observation, but we were unable to identify the information in the indicated line to make the suggested changes.
- Line 184 – subscript H2O2 . R: Change made.
- Line 208 – superscript mm2. R: Change made.
- Line 242 – This should actually be “We selected”. R: Change made.
- Line 409 – reference for “linked to the Th2 response?”. R: Change made.
- Line 411 – Regarding “caspase-3” as an apoptosis marker. R: We appreciate your contribution to our work, but we were unable to identify the information in the indicated line to make the suggested changes.
- Figure 5 – TNF-a in 5E is slightly cut off and the graph in 5A is cut off. R: Change made.
- Figure 5 legend – Include statistics description and what the values on the bar from -1 to 1 are. R: Change made.
Line 504 – Italicize Alouatta. R: Change made.
- Line 565/566 – Italicize Callithrix and Alouatta. R: Change made.
- Line 640 – Italicize. R: We appreciate your contribution to our work, but we were unable to identify the information in our text to make the suggested changes, as the aforementioned line is a reference.
- Supplementary table 1 – Dropped an “s” from species. R: Change made.
- Supplementary Figure 1 – Some labeling is not in English. R: Change made.
- Supplementary Table 6 – Typo: Interticial edema. R: Change made.
- Supplementary Table 7 – Typo: Satelitose. R: Change made.
- Supplementary Table 8 – Line numbers obscure table. R: Change made.
- Supplementary Table 8 – Typo: Satelitose. R: Change made.
- Supplementary Table 9 – Check column titles, i.e. P.ajustado. R: Change made.
